



# Efficient screening of groundwater head monitoring data for anthropogenic effects and measurement errors

Christian Lehr[1,2], Gunnar Lischeid[1,2]

[1] Leibniz Centre for Agricultural Landscape Research (ZALF), Müncheberg, Germany

[2] University of Potsdam, Institute of Environmental Science and Geography, Potsdam, Germany

Correspondence to: C. Lehr (lehr@zalf.de)





**Abstract**

Groundwater level data is monitored by environmental agencies to support sustainable use of groundwater resources. For this purpose a high spatial coverage of the monitoring networks and continuous monitoring in high temporal resolution is desired. This leads to large data sets that have to be quality checked and analysed to distinguish local anthropogenic influences from natural variability of the groundwater level dynamics at each well. Both technical problems with the

measurements as well as local anthropogenic influences can lead to local anomalies in the hydrographs. We suggest a fast and efficient screening method for identification of well-specific peculiarities in hydrographs of groundwater head monitoring networks. The only information required is a set of time series of groundwater head all measured at the same instants of time. For each well of the monitoring network a reference hydrograph is calculated, describing expected "normal" behaviour at the respective well as it is typical for the monitored region. The reference hydrograph is calculated by multiple

linear regression of the observed hydrograph with the "stable" principal components (PCs) of a principal component analysis of all groundwater head series of the network as predictor variables. The stable PCs are those PCs which were found in a random subsampling procedure to be rather insensitive to the specific selection of analysed observation wells, respectively complete series, and to the specific selection of measurement dates. Hence they can be considered to be representative for the monitored region in the respective period. The residuals of the reference hydrograph describe local deviations from the

normal behaviour. Peculiarities in the residuals allow to quality check the data for measurement errors and identify wells with possible anthropogenic influence. The approach was tested with 141 groundwater head series of the state authority groundwater monitoring network in northeast Germany covering the period from 1993 to 2013 in approximately weekly resolution.







**1 Introduction**

Sustainable management of groundwater resources aims to ensure that the abstraction of groundwater does not exceed groundwater recharge in the long run (e.g., Gleeson et al., 2012). This resonates in the water framework directive of the European Union which demands to achieve and maintain a good status of groundwater quantity, including the obligation to monitor the temporal development with sufficient spatial and temporal resolution to be able to distinguish between
anthropogenic influences and natural variability (EU, 2000).

In practice negative trends in the observed groundwater head time series serve often as first indication for anthropogenic effects. However, negative trends in hydrological systems do not necessarily imply anthropogenic influence, but might be rather indications of naturally occurring long term persistence (Hurst, 1951; Mandelbrot and Ness, 1968; Mandelbrot and Wallis, 1969) induced for example by the natural fluctuation of climatic drivers (e.g., Koutsoyiannis, 2006). Thus continuity
of hydrologic records for more than "just" a few decades is mandatory to incorporate this issue in water management (Hirsch, 2011). This holds the more for groundwater monitoring due to the much more pronounced filtering of short term fluctuations in the subsurface compared to precipitation, soil moisture or streamflow (Skøien et al., 2003).

Thus, sustainable management of groundwater resources requires a spatially differentiated and comprehensive monitoring of the groundwater level covering its temporal development continuously over decades which leads to large data sets. Checking
the data quality at the wells of a spatially comprehensive network can be very time consuming. Measurement errors as well as local anthropogenic influences lead to anomalies at individual sites. Thus, local anomalies in the hydrographs can serve as indication for both aspects. That requires a reference, either in the form of observed hydrographs measured at specific observation wells which are considered representative for undisturbed behaviour typical for the region (e.g., Winter et al., 2000; Gangopadhyay et al., 2001), or in form of some kind of modelled regional reference hydrograph. Here, we focus on
the second case.

One option is to apply a physically-based model for that purpose. Depending on the characteristics of the region and the level of considered details the amount of required data can be quite demanding (cf. discussion in Coppola et al., 2003). A non-exhaustive list of typical basic requirements includes climatological data to drive the model, data on the hydraulic properties of the subsurface which could comprise various aquifers, data on land use or time series of water abstraction, etc.
That information is required in a spatially distributed manner. In many cases, the monitoring effort, the effort to set up the model, the complexity of the model and the demand on data are substantial obstacles for environmental authorities or consultants at larger spatial scales.

Consequently, in practice some model parameters will serve during calibration as surrogate for missing data. For example, Wriedt (2017) calculated for each observation well in a monitoring network a "theoretical climatic hydrograph" by fitting the
monthly climatic water balance to the observed groundwater hydrograph using a damping and a translation factor. Here, the fitting of the damping and the translation factor compensated for missing information on different properties of the analysed groundwater system like different substrates, flow paths, etc.

Another option is to fit empirical models based on the relationships between easy to obtain independent variables and the observed water level. This has been performed for example by means of multiple linear regression (Hodgson, 1978),
artificial neural networks (Coulibaly et al., 2001; Coppola et al., 2003), the combination of an artificial neural network and a linear autoregressive model with exogenous input (Wunsch et al., 2018) or the combination of different methods from the fields of exploratory data analysis, information theory and machine learning (Sahoo et al., 2017). Those data-driven approaches make efficient use of the available data and are therefore recommended as a relatively cost and time efficient way to model groundwater level in areas with limited data (Hodgson, 1978; Coulibaly et al., 2001; Coppola et al., 2003).
Another benefit is that the respective models can easily be updated once new measurements or additional variables become available (Coppola et al., 2003).





Here, we suggest to derive local reference time series based on a Principal Component Analysis (PCA) of a set of monitored time series. This approach does not require any other data than the time series of water level all measured at the same instants of time. PCA is one of the most established, fastest and computationally cheapest statistical approaches to
summarize the spatiotemporal variability of a set of spatially distributed time series. Based on the linear correlation structure of the data set linearly independent principal components (PCs) are derived. Each PC is associated with one characteristic spatiotemporal pattern. Except from being fast and computationally very cheap, PCA is readily implemented in most of the common statistical software packages.

In analogy to climatology (e.g. Richman, 1986), the spatiotemporal patterns of the PCs have been used for long in hydrology
as a compact description of the dominant modes of hydrological variability in a region. Pioneering studies used it to describe dominant modes of streamflow variability in the European USSR (Smirnov, 1972, 1973), USA and Southern Canada (Bartlein, 1982), the USA alone (Lins, 1985a,b, 1997) and different regions of Sweden (Gottschalk, 1985). The leading PCs were used in combination with cluster analysis for the classification of streamflow (Hannah et al., 2000) and groundwater (Triki et al., 2014) hydrographs into groups with similar dynamics, which allowed for example to reduce the effort of
modelling of all the time series in a groundwater monitoring network to a few representative hydrographs (Upton and Jackson, 2011). These approaches have to be distinguished from using PCs for the identification of prevailing processes or functional relationships (e.g., Longuevergne et al., 2007; Lewandowski et al., 2009; Hohenbrink et al., 2016) which is beyond the scope of this paper.

To our knowledge, the application of PCA based approaches in the context of compact description of hydrological variability
focussed so far mainly on the leading PCs, that is, the main modes of hydrological variability on the scale of the analysed data set, like the main regional spatiotemporal patterns in a monitoring network. For example, Smirnov (1973) used the leading PCs for filtering small scale disturbances from the large scale patterns of long-period streamflow fluctuations in the European USSR. In groundwater monitoring, the leading PCs were used for example in the evaluation of groundwater monitoring networks to identify the observation wells which were most representative for the analysed region (Winter et al.,
2000; Gangopadhyay et al., 2001).

In this study, we used the leading principal components (PCs) of a set of groundwater head time series of a large scale monitoring network to decompose the observed hydrographs into a reference part and a residual part for each site separately. The reference hydrograph of an observation well describes the part of the observed hydrograph which is considered typical for the monitored region. It is determined by multiple linear regression of the observed hydrograph with the "stable" PCs as
predictor variables. The "stable" PCs are determined by comparing the results of a series of PCAs which were performed with different randomly selected subsets of the complete data set. The sub data sets were derived in the spatial domain as random selections of the observation wells, respectively complete series, and in the temporal domain as random selections of the measurement dates. Those PCs that turned out to be rather insensitive to the specific selection of analysed wells and measurement dates were defined as sufficiently stable and considered to be representative for the monitored region in the
analysed period. The residual part of the hydrograph depicts the local deviations from the normal behaviour at the respective observation well, which is then analysed for peculiarities pointing to technical problems or anthropogenic effects. Other applications of the reference hydrograph, like gap-filling in series which were not included in the PCA, are shortly discussed. In contrast to other PCA applications this approach does neither require any interpretation of the leading PCs nor any explicit spatial analysis. The approach was tested with 141 groundwater head time series of the authority's groundwater monitoring
network of the German Federal State of Mecklenburg-Vorpommern covering the period from 1993 to 2013 in approximately weekly resolution.



## 2 Data

### 2.1 Study region

The Federal State of Mecklenburg-Vorpommern is located in the Northeast of Germany (Figure 1) covering an area of 23,214 km² (Statistkportal, 2018). The hydrogeological structure in the area consists of several regional aquifer systems of Pleistocene origin that are considered to be in general hydraulically separated (Manhenke et al., 2001). The aquifers consist mainly of sandy and gravelly sediments, the intermediate aquitards mainly of till. The Pleistocene sediments usually overlie Tertiary sediments and can comprise more than 100 m. Within the Tertiary sediments the Rupelian clay layer hydraulically

separates the underlying saline Tertiary groundwater from the fresh groundwater (Manhenke et al., 2001). However, at some locations upwelling salty groundwater reaches the surface (compare LUNG (1984) and figure 2.11-6 in Martens and Wichmann (2007)), indicating that the hydraulic separation of the regional aquifer systems is not everywhere complete.

Groundwater in Mecklenburg-Vorpommern is monitored by the federal state office of Environment, Nature Conservation and Geology (LUNG). Monitoring comprised the uppermost three regional aquifer systems, but not all of them are

contiguous (Hennig et al., 2002). In addition to the regional aquifer systems, in some areas shallow local aquifers with an extent of usually a few km$^2$, occasionally more than 100 km$^2$, have been identified and are monitored as well (Hilgert and Hennig, 2017). These local aquifers are not strictly hydraulically decoupled from the regional aquifer systems, but their hydraulic connection is inhibited (Hilgert and Hennig, 2017). A mean annual groundwater recharge of +122.3 mm for the whole Mecklenburg-Vorpommern was estimated for the years 1971 to 2000 by Hilgert (2009) based on the work of Hennig

and Hilgert (2007). A map of the spatial distribution of the mean annual groundwater recharge is provided by the LUNG (LUNG, 2009). For the same period a mean annual temperature of 8.5 °C and a mean annual precipitation of 593 mm were observed by the German Weather Service (DWD) (DWD, 2018).

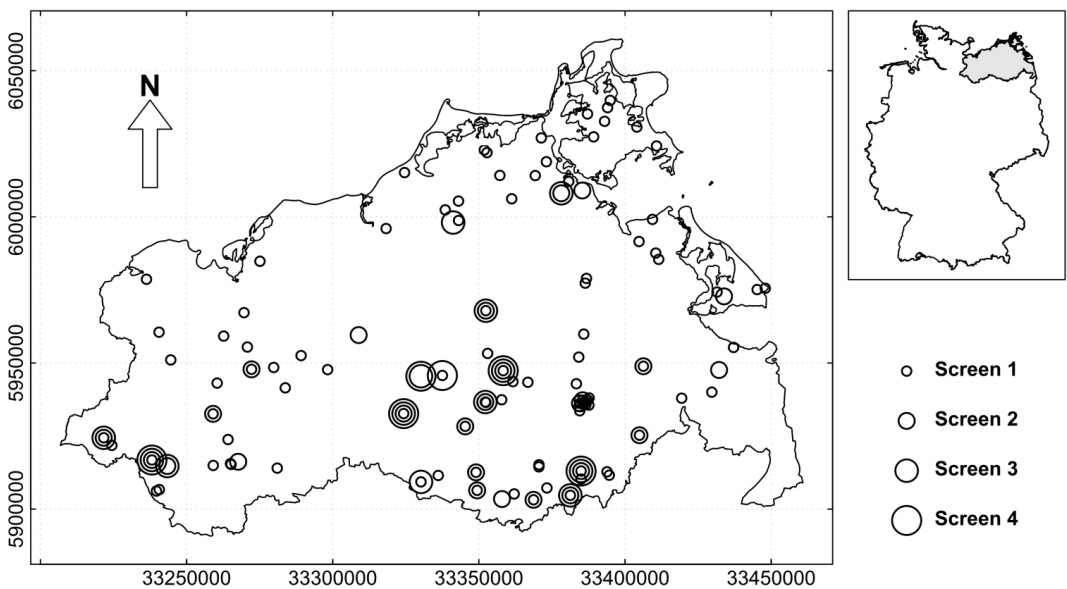

**Figure 1: Map of the study area and the selection of groundwater observation wells (N=141) in the federal state of Mecklenburg-Vorpommern (highlighted by grey shading in the inset map). The administrative borders of Germany and of the federal state of Mecklenburg-Vorpommern were obtained from GADM (2017).**





### 2.2 Groundwater head data and preprocessing

Groundwater level data was available for 583 groundwater observation wells of the groundwater monitoring network of the LUNG. In general, the data set covered a range of different monitoring periods. The mean time span from the first measurement till the last measurement of all observation wells was approximately 19.9 years. At some wells monitoring started as early as 1953. Especially in the beginning of the monitoring irregular measurement intervals were a rather common feature. In the last decades the state office aimed to take measurements consistently at least every 1st, 8th, 15th and

22th day of the month.

For this study, we selected 141 wells that covered the 20 years monitoring period from 1993-11-01 to 2013-10-22 (Figure 1). Seventeen sites with known anthropogenic effects were excluded beforehand. The dates of the readings of each of the 141 groundwater head series are shown in Figure S1. The variety of days between subsequent readings is shown for each series in Figure S2. The mean, minimum and maximum of all 141 series' mean measurement intervals were 5.3, 1.5 and 13.6 days.

The mean, minimum and maximum of all 141 series' maximum time gaps between subsequent measurement dates were 24.3, 10 and 88 days. All data gaps were linearly interpolated. Subsequently, regular quasi weekly time series were generated by selecting readings of the 1st, 8th, 15th and 22th of each month of the 20 years period. This resulted in a set of 141 groundwater head time series each with readings at the same 960 quasi weekly measurement dates (see the most left column of Figure S1). Thus, the last "quasi-week" exhibited different lengths for the different months.

The observation wells were irregularly distributed (Figure 1). This is a consequence of the mission of the state office to monitor possible anthropogenic influences on groundwater level, thus focusing on more densely populated areas. At 35 sites wells were screened at different depths. Distances between closest observation wells ranged from 0 km, at the sites with several wells, to 20.2 km, with a mean distance between closest wells of 3.4 km. At each site, the screens were numbered from the surface downwards (Figure 1). Different numbers of the screens do not necessarily imply different aquifers.

Comparing two observation wells of different sites, a higher screen number does not imply that the distance of the screen or of the water level to the surface is larger as well (Figure 2). The mean depth to groundwater, measured as distance of groundwater head to the cap of the well head during the observation period, ranged from 0.65 m to 30.96 m, with a mean of 6.30 m and a standard deviation of 5.36 m. The distribution of monitored mean depths to groundwater was heavily biased towards smaller depths (Figure 2). Due to the complex hydrogeological setting, the capture zones of the wells usually are not

known with sufficient detail.

As a rough approximation of the autocorrelation of the time series in spite of the not completely regular weekly sampling intervals the correlation of the series with itself shifted by one time step were assessed, yielding a correlation coefficient of r = 0.97 with a standard deviation of 0.04. The correlation in the spatial domain between the series of closest adjacent observation wells was substantially weaker with a mean correlation of r = 0.76 and exhibited substantially larger variability

with a standard deviation of 0.24.





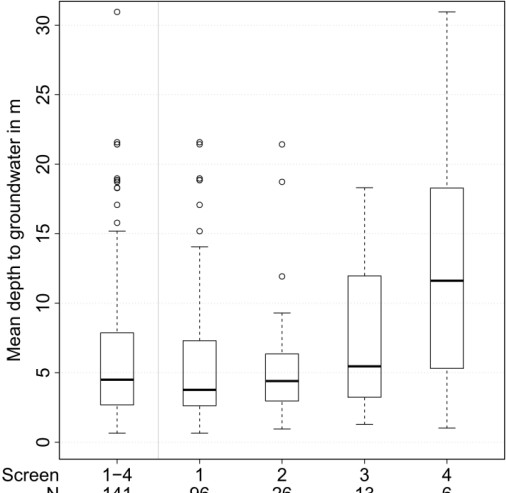

**Figure 2 Mean depths to groundwater at the observation wells in m of the complete set of series and separately for each screen number.**

**3 Methods**

**3.1 Principal component analysis**

The spatiotemporal variability in the groundwater head monitoring network was summarized with linearly independent principal components (PCs) determined by principal component analysis (PCA). The PCs are derived by an eigenvalue decomposition of the covariance matrix of the analysed set of variables, that is, the observed groundwater head series. To

achieve equal weighting of all series, we applied PCA to the z-scaled groundwater head series, thus each series was scaled to zero mean and standard deviation of one. This corresponds to performing PCA with the correlation matrix of the groundwater head series.

Each PC consists of an eigenvalue, eigenvector (loadings) and scores. The size of the eigenvalue of a PC in relation to the sum of all eigenvalues of all PCs corresponds to the share of variance in the data set that is assigned to that PC. The loadings

are the weights in the linear combination of the analysed variables, here the z-scaled groundwater head series, to calculate the scores of the PCs.

In this study, the analysed data set consisted of a set of time series all covering the same period and exhibiting the same dates. Thus, the scores of the PCs are times series with the same dates as the analysed time series. Please note, that this condition is a requirement for the suggested application of the reference hydrograph in this study. The Pearson correlation

coefficient was used to describe the relationships between the observed groundwater head time series and that of a selected PC. This yielded for each PC a characteristic spatial pattern of "occurrence" of the respective PC time series at the observation wells. The Pearson correlation values of this relationship correspond to the spatial pattern of the loadings of a PC multiplied with the square root of the eigenvalue of the respective PC. For better readability of the results we used here the Pearson correlation values for the maps of the loadings. Thus, each PC is associated with a characteristic temporal pattern

(time series of the scores) and spatial pattern (loadings).

We performed PCA with the function "prcomp" of the default package "stats" in R version 3.4.1 (R Core Team, 2017). For more details on PCA, please see Jolliffe (2002).





### 3.2 Stability of PCs

Being a data driven approach, the PCA results are depending on the selection of data. Thus, for the assessment of the typical regional behaviour it is crucial to use only those PCs which are rather insensitive to the specific selection of analysed observation wells, respectively complete series, and to the specific selection of measurement dates and hence can be considered to be representative for the monitored region in the studied period.

To assess the stability of spatial patterns of the PC loadings on the scale of the network, we performed 10,000 PCAs based

on random subsamples of the 960 quasi-weekly measurement dates and compared the PC loadings of the different runs. We calculated the squared Pearson correlation coefficient ($R^2$) of all combinations of loadings of PC 1 of the different PCA runs, all combinations of loadings of PC 2 of the different PCA runs, etc. for all PCs with eigenvalue larger one. This so called Kaiser criterion is a common threshold to select in case of PCA of z-scaled variables only those PCs which summarize more variance of the data set than one of the analysed variables (Jolliffe, 2002). The whole stability analysis was performed with

subsamples of 70% of all measurement dates. Analogously, we performed the stability analysis of temporal patterns of the PC scores with 10,000 PCAs each based on random selections of 70% of the 141 complete groundwater head series. We considered only those PCs as stable of which the correlations of both, the spatial as well as the temporal patterns, exhibited a median $R^2 > 0.9$.

### 3.3 Well specific reference hydrograph and residuals


The focus of this study is to present an approach to quickly screen all groundwater head series in a comprehensive monitoring network for problems with data quality and anthropogenic effects. For this purpose each observed hydrograph is decomposed into a "normal" part describing the behaviour as it is typical for the monitored region and the well specific deviations from it. The "normal" behaviour at each observation well was estimated as well specific reference hydrograph by

multiple linear regression of the observed series with the time series of the scores of the stable PCs. The residuals from the regression (residuals), which is the part of the series that has not been assigned to the reference hydrograph describe the local deviations from the normal behaviour at each observation well.

In contrast to many other approaches, it is not required that the residuals are white noise or alike. Possible systematic structures in the residuals like trends, seasonal patterns, sudden shifts or distinct periods of deviations at a specific well are

not captured by the stable PCs and therefore indicative that the respective pattern is not representative for the whole data set, but a local peculiarity instead. The squared linear correlation coefficient $R^2$ of the observed versus the respective reference hydrographs was used for a first assessment which observation wells exhibit rather normal behaviour and which do not. Analysing the temporal pattern of the residuals in the context of expert knowledge and other information available for the respective observation wells can then be used to derive hypotheses on the causing drivers.


### 4 Results

### 4.1 Stability of PCs

The first ten PCs of the PCA of the complete data set exhibited eigenvalues larger one. Thus in the following we present the results of the first ten PCs only. The comparison of the spatial patterns of the PCAs which were performed with the sub

datasets based on the random subsampling of the measurement dates is summarized in Figure 3a. In general, the median correlation between the loadings of the PCs of the different PCAs was decreasing and the variability of correlation between the loadings was increasing with increasing rank of the PCs (Figure 3b). All PCs except the 8[th] and the 10[th] exhibited notably stable spatial patterns with a median correlation of $R^2 > 0.9$.





The comparison of the temporal patterns of the PCAs which were performed with the sub datasets based on the random

selections of observation wells, respectively complete series, is summarized in Figure 3b. Generally, with increasing rank of the PCs the median correlation between the scores of the PCs of the different PCAs was decreasing and the variability of correlation between the scores was increasing (Figure 3b). Only the first four PCs exhibited notably stable temporal patterns with a median correlation of $R^2 > 0.9$.

Accordingly, the first four PCs were considered stable on the scale of the network, accounting for 80.8% of the observed

variance in the groundwater head series. The variance assigned was for PC 1: 48.3%, PC 2: 17.2%, PC 3: 9.5% and PC 4: 5.8%. The assigned temporal and spatial patterns of the complete data set with all 141 series are shown in Figure 4 and Figure 5, respectively. In the following the analysis is restricted to these four stable PCs.

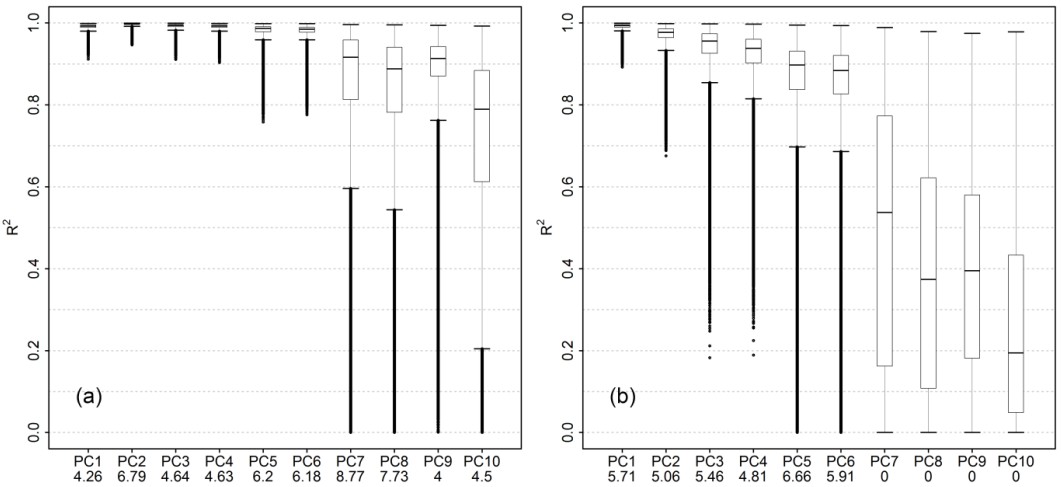

**Figure 3: Comparison of the PCA results of the stability analysis. (a) Correlation of loadings based on the random subsampling of the measurement dates to assess the stability of spatial patterns. (b) Correlation of scores based on the random selection of observation wells, respectively complete series, to assess the stability of temporal patterns. The boxes indicate the quartiles, the whiskers all dates which are within the range of the first quartile - 1.5\* the interquartile range, respectively the third quartile + 1.5\* the interquartile range. Percentage of values outside the whiskers of a boxplot is given in the labelling of the x-axis for each**

**PC.**





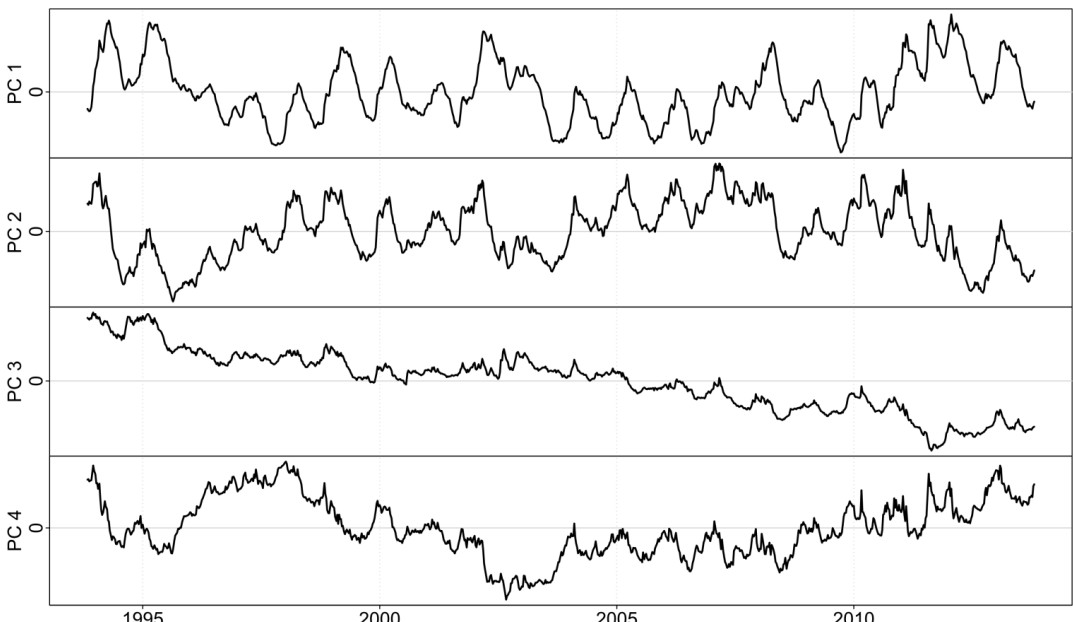

**Figure 4: Time series of scores of the stable PCs 1 to 4.**

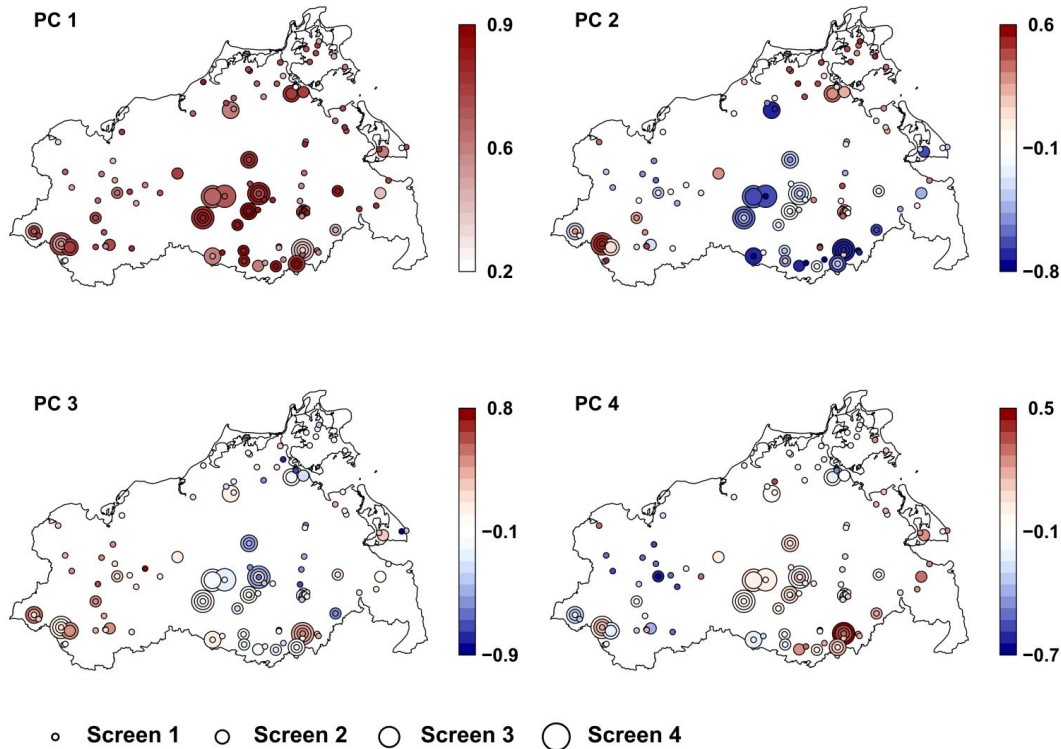

**Figure 5: Spatial patterns of loadings of the stable PCs 1 to 4.**





### 4.2 Well specific reference hydrograph and residuals

At each well, the residuals of the reference hydrograph were checked for peculiarities. We present here two examples. At well Deven (Figure 6) the observed hydrograph was in general very well represented by the well specific reference

hydrograph ($R^2 = 0.84$). The mean depth to groundwater was 9.52 m. A single period stuck out in the plot of the residuals. At the end of October 1998 the residuals showed a sudden shift of approximately 17 cm towards higher water level, followed by a similar sudden shift "back to the old level" in December 1999. Those shifts were neither outstanding in the observed time series itself nor in the reference hydrograph.

Observation well Neubrandenburg UP exhibited in general a relatively good fit of observed series and reference hydrograph

($R^2 = 0.77$) with the exception of two outstanding periods in 1997-1998 and 2007-2008, a series of minor deviations before 1997 and another relatively strong deviation in 2011 (Figure 7b and d). The mean depth to groundwater was 4.61 m. For comparison we considered the close-by observation well NB-Hotel Vier Tore approximately 600 m further south which was formerly excluded from the PCA due to known anthropogenic influence (section 2.2). The mean distance of well head to groundwater level was 3.74 m. Here we calculated the reference hydrograph in the same manner as multiple linear regression

of the observed hydrograph with PCs 1–4. Again the observed series exhibited in general a relatively good fit with the reference hydrograph (Figure 7c and d). However, compared to observation well Neubrandenburg UP the two periods of strong deviation in 1997-1998 and 2007-2008 were more pronounced in the residuals and, in contrast to observation well Neubrandenburg UP, clearly visible in the observed series as well. This was reflected in a substantially weaker correlation between the observed series and the reference hydrograph ($R^2 = 0.46$).


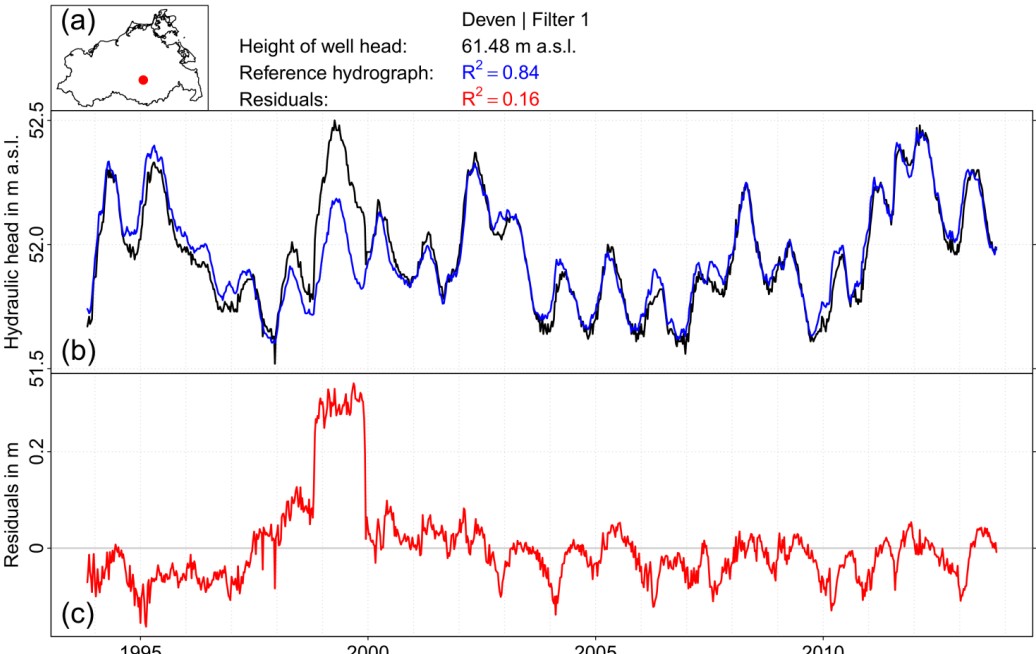

**Figure 6: (a) Location of the well, information on the observation well and correlation of the observed series with the reference hydrograph and with the residuals. (b) Time series of hydraulic head (black) and the reference hydrograph (blue) of well Deven. (c) Time series of residuals.**



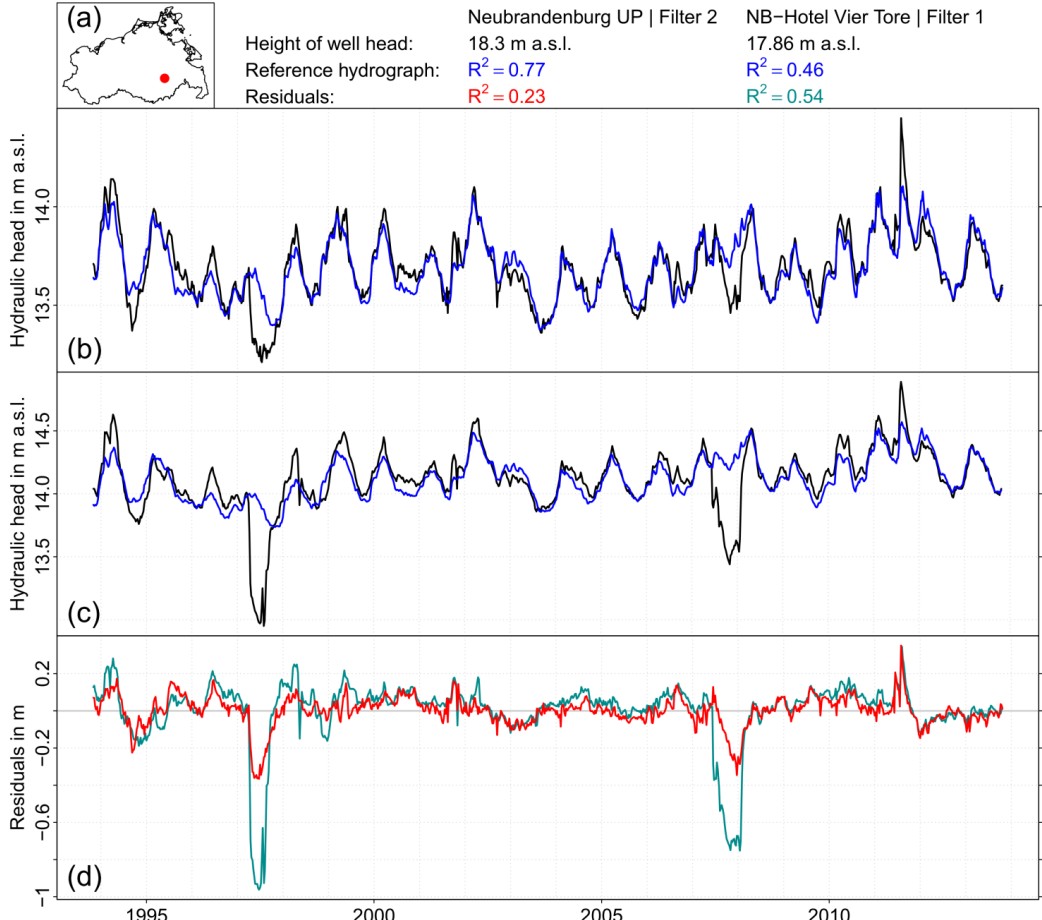

**Figure 7 (a) Location of the wells, information on the observation wells and correlation of the observed series with the reference hydrographs and with the residuals. Time series of hydraulic head (black) and the reference hydrographs (blue) of (b) Neubrandenburg UP and (c) NB-Hotel Vier Tore. (d) Time series of the residuals of Neubrandenburg UP (red) and of NB-Hotel Vier Tore (dark cyan).**

## 5 Discussion

### 5.1 Stability of PCs

To select only those PCs which are representative for the monitored region in the analysed period a series of PCAs were performed based on randomly selected subsets of the complete data set to identify the stable PCs, that is, those PCs of which the assigned spatial patterns (loadings) and temporal patterns (scores) were rather insensitive to the selection of analysed observation wells and measurement dates. Only the stable PCs were considered for the further analysis.

Earlier studies which used PCA to summarize hydrological variability in a region analysed the stability of their results in a similar way (Smirnov, 1973; Lins 1985a). Those attempts were limited to the comparison of a few PCAs, respectively a few different configurations of the data set. The correlation analysis in this study extended the assessment of stability of PCs towards random selections of the analysed data. An important difference to the aforementioned studies is that the selections did not occur en bloc but randomly among all observation wells and all measurement dates.





The comparison of the results of the 10,000 variants of PCA revealed that the stability of individual PCs decreased in general with decreasing rank of the PCs (Figure 3). Clear differences between the stability of temporal and spatial patterns of the PCs were observed. PCA results were more sensitive to changes in the selection of considered observation wells (Figure 3b)

than to changes in the selection of measurement dates (Figure 3a). This likely reflects the stronger mean correlation between subsequent observations at the sites (autocorrelation) compared to the mean correlation of the series of closest adjacent sites (spatial domain) (section 2.2). The first four PCs were found to be stable (Figure 3). This gave some confidence that their spatial and temporal patterns were indeed characteristic for the monitored region in the analysed period and not restricted to the specific selection of observation wells or measurement dates. Compared to the established Kaiser criterion (section 3.1)

the number of considered PCs was reduced from ten to four.

### 5.2 Well specific reference hydrographs and residuals

To account for well specific peculiarities we examined the time series of residuals of the reference hydrographs (for example Figure 6 and Figure 7). These peculiarities might be of different origin. First of all they might be caused merely by technical

problems. For example we interpreted the single period in 1999 which sticks out in the plot of the residuals of observation well Deven as a step-wise shift of the logger (Figure 6c). This shift was in phase with the seasonal pattern of the observed series and was therefore not obvious from the visual inspection of the observed series alone, while it was clearly visible in the residuals.

An example for well-specific peculiarities in the residuals due to local anthropogenic influence is given in Figure 7.

Observation well NB-Hotel Vier Tore was excluded from the PCA because its hydrograph was known to be influenced by the lowering of groundwater level due to construction works of underground car parks in 1997-1998 and 2007-2008 approximately 100 and 200 m apart, respectively. While this influence was clearly visible in the observed series at that observation well (Figure 7c), it was not obvious in the observed series at observation well Neubrandenburg UP, especially not the second deviation in 2007-2008 (Figure 7b). This is most probably because Neubrandenburg UP was further away

from both construction sites, namely approximately 400 m each. However, in the residuals both periods became clearly visible for both observation wells although to different degrees at the two wells (Figure 7d).

Such clear assignment of anthropogenic influence to a local deviation from the regional behaviour is only possible if the scale of the respective effect is rather local in comparison to the scale of the monitoring network as a whole, and the scale of spatiotemporal resolution of the network in particular. The latter enables a distinct localization of the influence. An

anthropogenic effect which induces similar groundwater head dynamics at a substantial amount of the series of the data set would be incorporated in the leadings PCs (Wriedt et al., 2017), and thus would affect the reference hydrographs. However, such an effect is hardly likely. Moreover, the presented approach does not differentiate anthropogenic from "natural" effects per se. Rather, it decomposes the time series into regional patterns which can be assigned to many or all time series of the data set and local patterns which are restricted to a few or single sites. Another restriction is that PCA considers only

temporal patterns of groundwater head, but ignores the absolute values. Thus, it does not allow any inferences whether the observed groundwater head on average is higher or lower than it would be under natural conditions.

The spatial clustering of observation wells indicated a spatial bias of the monitoring (Figure 1). It reflected the focus of the monitoring on anthropogenic water use, for example close to settlements, towns, etc., which is prerequisite for sustainable water management. Because all the series were equally weighted by z-scaling (section 3.1), the derived PCs and

consequently the determined normal behaviour were biased towards areas with higher density of observation wells (Karl et al., 1982). This should be considered for any interpretation of the reference hydrographs, respectively the general behaviour of the groundwater level in the region, as well as the local deviations. However, in this study, the stable PCs used for the



estimation of the regional behaviour turned out to be rather robust with respect to the selection of observation wells (section 4.1), suggesting that the results were not primarily determined by the local clustering of the observation wells.

In general, the reference hydrograph is a relatively good-natured and robust PCA application for mainly two reasons. First, the selection of the considered PCs is transparent and reproducible. The approach prevents the consideration of PCs which exhibit pronounced instability of the associated spatial and temporal patterns, for example PCs with "degenerate eigenvalues", that is eigenvalues which are indistinguishable within their range of uncertainties (Hannachi et al., 2007). Second, not the single PCs are used, but the combination of the stable PCs. Thus, it is not necessary to interpret single PCs as

drivers of groundwater head variability, distinct processes or alike. Consequently, describing the regional behaviour with the reference hydrograph is also applicable in cases in which the interpretation of single PCs is severely hampered, for example if the associated spatial patterns of the PCs mainly reflect the shape of the analysed domain (domain shape dependence) (Buell, 1975, 1979; Richman, 1986, 1993).

Some PCA applications involve rotation of the considered PCs to achieve more simple structures, respectively more

localized spatial patterns, which might support the interpretation of single PCs (Lins, 1985b; Richman, 1986; Jolliffe, 1987; Jolliffe, 2002; Hannachi et al., 2007). It is possible to combine such applications with the reference hydrograph application. For the suggested screening application rotation of the PCs does not change the results, as long as the rotation is performed with all stable PCs or only with a subset of the stable PCs. Than the reference hydrographs, respectively the residuals, are the same whether they are calculated from the rotated or the un-rotated PCs. Concerning the reference hydrographs the decisive

question is which PCs are included in the calculation.

The presented approach uses the spatiotemporal variability in a large set of groundwater head series to determine individual reference hydrographs for each observation well. Thus, it does neither require the identification of clusters of similar wells or single reference wells, nor assumptions on the catchments of the wells, hydraulic connection between the wells, etc. This is in contrast to approaches which select some of the monitored wells as reference observation wells being representative for

the whole monitoring network or for subgroups or sub-regions only. For example other PCA applications used the clustering of observation wells in the scatterplot of loadings of PC 1 versus PC 2 to identify "index" wells for each cluster (Winter al., 2000) or applied PCA directly to subgroups of a monitoring network, which were determined based on an estimation of the physical relatedness of the observation wells before, to identify firstly the "principal wells" of the subgroups and subsequently the most representative wells for the whole network by ranking all wells according to their number of

occurrences as principal well (Gangopadhyay et al. 2001).

However, despite the different approaches of how to determine the most representative observation wells it has to be considered that observation wells which are of little representative value for the whole monitoring network might be of high informative value for example with respect to anthropogenic influence specific to single observation wells. In contrast to the selection of single wells which are considered especially representative or atypical for the network, the suggested approach

in this study enables to order all wells in a network quickly according to their representativeness and yields for each well an estimation of the local well-specific behaviour.

### 5.3 Other applications

In addition to the presented applications in this study, other applications of the reference hydrographs and residuals are

possible. One option is to use the series of the stable PCs (scores) as predictor variables in a linear regression to fill gaps in hydraulic head series which were not analysed with the PCA but which exhibit some overlap with the monitoring period covered by the PCA. For example the reference hydrograph of observation well NB-Hotel Vier Tore (Figure 7c) was calculated although the groundwater head series was not part of the PCA. In this study it was merely used for identification of the influence of the construction works, but it could be used to replace the periods which were influenced by the





construction works with an estimation of unaffected groundwater head dynamics as well (Figure 7c). In the spatial domain, calculating the Pearson correlation of excluded series with the stable PCs for limited overlapping time periods can be used to extend the spatial coverage of the loadings maps. For both applications, the random subsampling analysis can be used to estimate how many missing values in the series might be tolerable. For this data set we would be confident to consider series with up to 30% missing values during the 20 years-period. The maximum time gap in this study was 88 days (section 2.2)

and most of the time gaps were substantially smaller (Figures S1 and S2). Thus, the results are most likely less stable for a few long gaps compared to numerous but short gaps.

     Another application is to identify distinct reference observation wells by selecting those observation wells at which the correlation between the reference hydrograph and the observed series is above a certain threshold, for example $R^2 > 0.9$. Those would be considered as most representative observation wells for the whole monitoring network. In case the period

covered by the series of the reference observation wells exceeds the period covered by the PCA, they can be used as predictor variables in a linear regression to extrapolate the series of the stable PCs scores.

     The methods to extend the spatial and temporal coverage of the PCs should be handled with care. However, because only the stable PCs were used, there should be no major bias, as long as the extension is performed only for some years or small numbers of additional series. If new data are available, that cover a larger area or a longer period, it is in general preferable

to perform a new PCA with all available data to account for systematic changes in the temporal dynamics of the analysed groundwater system, and systematic changes in the monitoring network geometry and spatial distribution of the observation wells.

     The detection of changes in characteristic temporal patterns (scores) and their occurrence in the monitoring network (maps of loadings) between different observation periods is another application of the reference hydrographs and their residuals. It has

to be noted that a direct comparison of the temporal patterns of two periods can only be performed for temporally overlapping periods. For non-overlapping periods, a comparison is restricted to general time series characteristics like for example the timing or amplitude of a seasonal pattern.

## 6 Conclusions

     We suggested and tested a PCA based approach for the fast and efficient screening of time series of groundwater head of a

monitoring network for technical problems and anthropogenic effects. Here, each observed series is decomposed in two parts. The reference hydrograph part describes "normal" behaviour as it is typical for the respective study region. The residual part describes local deviations from the normal behaviour. Peculiarities in the residuals serve as indication for technical problems or anthropogenic influence. The reference hydrograph at each well is calculated by multiple linear regression of the observed hydrograph with the stable PCs. The stable PCs are those which are considered representative for

the monitored region in the analysed period. They were identified in a random subsampling procedure as those PCs of which the associated spatial and temporal patterns are relatively insensitive to the specific selection of analysed data points in space and time, here the observation wells and measurement dates. The approach to determine the stable PCs is transparent and reproducible.

     The application of the reference hydrographs and their residuals proved to be a straightforward way to quality check the data

and to identify candidates for local anthropogenic influence. Both applications are actually an interpretation of the temporal dynamics of local anomalies in the observed groundwater head series. In contrast to other approaches the identification of those local anomalies is based on the correlation among the observed groundwater head series only and not based on physical models or empirical relationships with any predictors of groundwater head dynamics. It also does not require an interpretation of single PCs as distinct physical processes or functional relationships. This limitation with respect to direct

physical interpretation of the results brings with it the benefit that the only information required are series of (groundwater)



hydraulic head readings all measured at the same instants of time. Other suggested applications for the stable PCs are for example data-driven gap-filling in the observed series, spatial and temporal extrapolation of the reference hydrographs or the identification of distinct reference observation wells.

The computational demand is very low. Time series of the well specific deviations from normal behaviour (residuals) are
easily derived and enable a fast screening for well-specific peculiarities. In monitoring practice, the well-specific residuals can be used to distribute resources according to the "normality" of an observation well, in particular to support the decision which observation wells, respectively which series, should be investigated in more detail. Furthermore it can be used to categorize the deviations from the normal behaviour. Thus, we recommend the presented approach as a fast screening tool for the assessment of comprehensive groundwater monitoring networks.


## Acknowledgements

The first author received funding by the State Office of Environment, Nature Conservation and Geology (LUNG) of the federal state of Mecklenburg-Vorpommern. We thank Beate Schwerdtfeger and Heike Handke from LUNG for the initialisation of the project, the provision of the data and the productive collaboration throughout the project, Christoph
Merz, Philipp Rauneker and Katharina Brüser from Leibniz Centre for Agricultural Landscape Research (ZALF), Marcus Fahle from the Federal Institute for Geosciences and Natural Resources (BGR) and Tobias Hohenbrink from the Technical University of Berlin (TU Berlin) for the fruitful discussions, Berry Boessenkool for the inspiration for Fig. S1, Klaus-Dieter Fichte, Toralf Henke and Michael Höft from the State Office of Agriculture and Environment Vorpommern (STALUP), as well as Hanjo Polzin and Peter Schuldt from the Mining Office of Mecklenburg-Vorpommern in Stralsund, Konrad Höppner
from CEMEX Germany, Torsten Abraham from Hydro-Geo-Consult GmbH and Jacob Möhring from LUNG for expert knowledge on different observation wells, Heiko Hennig from Umweltplan Greifswald and Toralf Hilgert from HGNord for information on former projects with similar data sets in the study region, and Andreas Mitschard from LUNG for the continuous technical support.

**Data availability**

Up to date groundwater head series of the observation wells used in this study are provided on request by the State Office of Environment, Nature Conservation and Geology (LUNG), Güstrow, Germany.

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
