# Peer review of "Efficient screening of groundwater head monitoring data for anthropogenic effects and measurement errors"

_Hydrology and Earth System Sciences, 2019_

## Referee Comment (RC1) · Anonymous Referee #1 · 5 Aug 2019

**Efficient screening of groundwater head monitoring data for anthropogenic effects and measurement errors**

*General review comments*

The study applied Principal Component Analysis to identify anomalies in observed groundwater level time-series, as a means to screen a monitoring network for anthropogenic effects and measurement errors. This was achieved by constructing a "reference hydrograph" for each observational record, via multiple linear regression of the stable principal components derived from the PCA analysis of the 141 sites in the network. The differences (residuals) between each reference hydrograph and the observed hydrograph were used to identify periods of erroneous, or anthropogenically impacted, measurement. The presentation of the methodology is reasonable and the approach could be of significant value to others involved in the quality assurance of monitoring network data, if it is "efficient" as suggested by the title. The paper would make a useful contribution to the literature. However, the problem is that the authors do not provide sufficient analysis of the application of the methodology to the *network*, i.e. the full set of 141 groundwater level hydrographs, to convince the reader of this. Rather, they just analyse two groundwater level time-series to provide examples of where erroneous data has been found, one of which has a neighbouring hydrograph where pumping impacts are known about. For example, and with reference to the following two sentences, there is no discussion of how records containing anomalies were identified after the residual time-series were calculated.

> Line 226: "The focus of this study was to present an approach to quickly screen **all** groundwater head series in a comprehensive monitoring network for problems with data quality and anthropogenic effects."

> Line 238: "Analysing the temporal pattern of the residuals in the context of expert knowledge and other information available for the respective observation wells can then be used to derive hypotheses on the causing drivers."

National monitoring network I know of, and use, may have >10,000 records in them. The question I am left with is how this methodology could be such large networks, as this would be of great value. Currently, the analysis presented underwhelms as it just shows a couple of examples, which were presumably selected because the anomalies they contain were easy to spot? I am left not knowing how much "expert judgement" was used in identifying anomalies in the residual series, as there really is no description of how the analysis of the "temporal pattern of the residuals" was performed. I think this should be addressed before publication of the paper.

A final general comment is that the English grammar could/should be improved; this is principally due to the use of sentences which are very long, and which contain many clauses. If these can be split and simplified I'm sure that the paper will become much more readable.

*Specific review comments*

| Line | Comment |
|------|---------|
| - | Some corrections to the grammar/English have been made and tracked in the attached document. The PDF was opened in Word, which is not perfect, but should allow the authors to see the suggested edits easily. |
| 11 | "a high spatial coverage of the monitoring networks"

This doesn't sound right. Perhaps you mean a high density of monitoring points within a network? It could be read to mean that one monitoring network should cover a large area. |
| 73-81 | This brief review of "empirical" models is somewhat unbalanced and therefore weak. It provides examples three specific methods, for which the references are quite old, and provides one reference citing the whole of data analysis, information theory and machine learning. I think a better brief review of methods that would be of more use to the reader could be written within a similar amount of space.

Here are a few examples of other / recent approaches?

https://www.sciencedirect.com/science/article/pii/S002216941930006X

https://ngwa.onlinelibrary.wiley.com/doi/abs/10.1111/j.1745-6584.2007.00382.x https://www.sciencedirect.com/science/article/pii/S0022169416303651 |
| 125-132 | The hydrogeological structure is not clearly described. You say "several regional aquifer systems". Does this mean horizontally separated aquifers, or do you mean a series of aquifers within a vertical stratigraphical sequence that extends across the study area. This is just not clear and should be explained more systematically, preferably with a hydrogeological map (unless it is a horizontally homogeneous, but vertically stratified system). |
| 134 | Again, here you say "uppermost three regional aquifer systems". But the use of the word "system" could be taken to mean that you are combining a series of discrete aquifer (vertical units?).  It's just difficult to understand what the system is and what the aquifers are. |
| 140 | Can you summarise what the recharge distribution is like, otherwise it is of not much interest to the reader. Is there a recharge gradient. Are there any spatial discontinuities in recharge that could influence the PCAs? |
| 141 | The sentence beginning "For the same period" has no direct relevance to the analysis, is of not much interest on its own, and could be deleted. |
| 152 | "time span from the first measurement till the last measurement of all observation wells"

This is verbose. What about "the mean length of measurement record was 19.9 years" |
| 153 | "In the last decades" is vague. What about "Since approximately 20##.." You have the data so can easily estimate this. |
| 176-180 | It is not really that sensible or informative to calculate autocorrelation at lag 1, as of course, one would expect a number like 0.97. What about looking at, for each time-series, the duration over which (de-seasonalised) levels show autocorrelation i.e. the correlogram values are greater than the error bounds. Does the degree of autocorrelation across the region show any spatial pattern that relates to the PCAs? |

|  | Given that the distance varies between the boreholes, stating the mean correlation in space is also not very informative. Why not show a variogram. |
|---|---|
| 210 | Regarding the sentence beginning "Thus, for the assessment". There are a number of sentences that are long and that contain the word "respectively". This is generally used poorly, making the sentences difficult to read. I would split all of these sentences into two, making each simple and clear, and stop using "respectively". |
| 229 | "was estimated as well specific reference hydrograph" is not grammatically correct and makes the sentence difficult to understand/read. |
| 281 | "a series of minor deviations before 1997 and another relatively strong deviation in 2011 (Figure 7b and d)". It is difficult to make this out. Can you add minor tick marks to the x axis on both Figure 6 and 7? The "relatively strong deviation" looks like it is later than 2011. |
| Figure 6&7 | The use of colour is not great, and will be undiscernible for those with (common) red-green colour blindness. Why not use dotted and dashed lines, and please add a legend to each plot, which will make it quicker to differentiate between the reference and observed hydrographs.  I think the eight lines in the very top right of the figure should be moved to their relevant position in each plot (box) |
| 368 | "Than the reference hydrographs, respectively the residuals, are the same whether they are calculated from the rotated or the un-rotated PCs." This a poorly constructed sentence and therefore difficult to understand. |
| 419-428 | The first part of the conclusion is just repetition of the methodology, and should be deleted, as it is not of mush interest here. There is generally quite a lot of repetition throughout the manuscript relating to the methodology, for example the first sentence of section 5.2. |
| Figure S1 | This figure doesn't provide the reader with much (if any) useful information/knowledge about the dataset. Much better would be one/some plots summarising some of the statistical properties of the observations e.g. density plots summarising changes in the frequency of measurement over time would provide more useful information. |

c Author(s) 2019. CC BY 4.0 License.

[revised manuscript text omitted]

---

## Referee Comment (RC2) · Anonymous Referee #2 · 18 Sep 2019

The paper addresses time series analyses for piezometric heads measured on different wells belonging to a dense network. The topic of the paper is suitable for HESS. The analyses are performed with a known statistical method (Principal Component). The novelty of the paper is the application of this method to piezometric heads chronicle to detect peculiarities in hydrographs of groundwater head. 141 groundwater head time series were selected from 583 wells. The selection criteria are for me unclear. Were they selected because they reach different aquifers? Using a first screening using statistical criteria? Furthermore, piezometric fluctuations with known anthropogenic influences are excluded from the PCA. Does it mean that the method detects only "minor" (not obvious from the visual inspection) peculiarities? Groundwater heads are

usually depending on groundwater recharge, the thickness of the unsaturated zone, exchange with rivers that can have different time characteristics. Under such very different conditions, the computation of the reference hydrograph is not obvious and need some more details (see §3.3). Could you provide some more details for two very different time series and how the PCs included in the calculation are chosen? Mean depth to the ground surface are analyzed. Therefore, systematic errors due to the vertical coordinate of the well reference cannot be detected. Moreover, the provided examples show time limited peculiarities. Is the method suitable to detect long term peculiarities like drifts?

L319 - Reference to Kaiser criterion is 3.2 and not 3.1

---

## Author Comment (AC5) · 3 Oct 2019

During the work on the comments of Referee 2 we discovered one detail we would like to change to improve the clarity of the used terms at the beginning of the manuscript. In the introduction in lines 108+109 we extended the wording similar to lines 18+19 in the abstract:

"The reference hydrograph of an observation well describes the expected "normal" behaviour at that well, i.e. the part of the observed hydrograph which is considered typical for the monitored region."

---

## Author Response (AR1)

**EDITOR:**

Editor Decision:

Reconsider after major revisions (further review by editor and referees) (03 Nov 2019) by Alberto Guadagnini

Comments to the Author:

Major critical points have emerged during the review process, as clearly stated by the constructive comments of the Reviewers, both reviewers being very critical on some key aspects. These, together with the replies proposed by the Authors suggest that the manuscript requires a set of major revisions prior to being considered again. It is not my intention to disregard any of the points raised by the Reviewers and it should be clear that there is no guarantee that the manuscript be accepted unless all of the Reviewers are unambiguously satisfied.

**AUTHORS:**

Dear Alberto Guadagnini,

Thank you very much to reconsider the revised version of our manuscript.

Please find below our detailed replies to all comments of the Referees, our two author comments and a marked-up version of the manuscript showing the applied changes. The marked-up version of the manuscript includes the changes we suggested in our replies to the comments of the Referees as well as formal changes like corrections of spelling. For our replies to the Referees we marked the unaltered replies from the 3$^{rd}$ of October 2019 with blue font color. New or modified replies are marked with red font color.

We thank once again both Referees for their time, work and the constructive comments. The comments helped us a lot to improve in general the quality, and in particular the readability and clarity, of the manuscript. We hope that this corresponds to the assessment of both Referees and that we could clarify the raised issues to their satisfaction.

Best regards

Christian Lehr

(on behalf of the authors)

**Anonymous Referee 1**

**AUTHORS:** We thank Referee 1 for her/his time, work and the constructive comments. This helped to improve and clarify our manuscript. For the review process we reply to the general review comments (GRC) in three subitems. For the specific review comments (SRC) we changed the tabular form to running text and added numbering.

Unaltered responses from the 3rd of October 2019 are marked with blue font color. New or modified responses are marked with red font color.
* * *
*General review comments (GRC):*

**REFEREE GRC #1:** The study applied Principal Component Analysis to identify anomalies in observed groundwater level time-series, as a means to screen a monitoring network for anthropogenic effects and measurement errors. This was achieved by constructing a "reference hydrograph" for each observational record, via multiple linear regression of the stable principal components derived from the PCA analysis of the 141 sites in the network. The differences (residuals) between each reference hydrograph and the observed hydrograph were used to identify periods of erroneous, or anthropogenically impacted, measurement. The presentation of the methodology is reasonable and the approach could be of significant value to others involved in the quality assurance of monitoring network data, if it is "efficient" as suggested by the title. The paper would make a useful contribution to the literature.

**AUTHORS:** We thank the Referee for the generally positive valuation of our study.

**REFEREE GRC #2:** However, the problem is that the authors do not provide sufficient analysis of the application of the methodology to the *network*, i.e. the full set of 141 groundwater level hydrographs, to convince the reader of this. Rather, they just analyse two groundwater level timeseries to provide examples of where erroneous data has been found, one of which has a neighbouring hydrograph where pumping impacts are known about. For example, and with reference to the following two sentences, there is no discussion of how records containing anomalies were identified after the residual time-series were calculated.

> Line 226: "The focus of this study was to present an approach to quickly screen **all** groundwater head series in a comprehensive monitoring network for problems with data quality and anthropogenic effects."

> Line 238: "Analysing the temporal pattern of the residuals in the context of expert knowledge and other information available for the respective observation wells can then be used to derive hypotheses on the causing drivers."

National monitoring network I know of, and use, may have >10,000 records in them. The question I am left with is how this methodology could be such large networks, as this would be of great value. Currently, the analysis presented underwhelms as it just shows a couple of examples, which were presumably selected because the anomalies they contain were easy to spot? I am left not knowing

how much "expert judgement" was used in identifying anomalies in the residual series, as there really is no description of how the analysis of the "temporal pattern of the residuals" was performed. I think this should be addressed before publication of the paper.

**AUTHORS:** The focus of the study was to present the approach, rather than presenting a detailed analysis of the full set of 141 groundwater level hydrographs. Therefore, we decided to pick two illustrative examples of the application of the reference hydrograph. The selected examples are also in this regard illustrative that the anomalies are "easy to spot" in the residuals of the reference hydrographs, but not in the observed hydrographs themselves.

In fact, the complete data set was quality checked before the analysis by the state agency. The analysed data set comprised only those groundwater head series which were considered by the state agency (LUNG) to be without any anthropogenic effects or measurement errors.

In the project, the identification of anthropogenic effects and measurement errors was carried out with two steps. Firstly, the wells were ranked according to the strength of correlation of the observed and the respective reference hydrographs. Those wells at which the observed series were not well represented by the respective reference hydrograph, i.e. those which exhibited low $R^2$ with the reference hydrograph and high $R^2$ with the residuals, were considered the most promising candidates to exhibit anthropogenic effects or measurement errors. Secondly, in the order of this ranking, we visually examined the residuals for peculiarities, for example step changes, trends, periods of strong deviations, etc. Those were than discussed with the experts from the state agency.

Basically the approach enabled to focus the knowledge on the local conditions of the experts to distinct wells and periods of time. To our understanding this is what should be accomplished by a screening tool.

Other than visual inspection, time series tools like filters can be used to identify for example step changes or trends in the residuals. Especially for larger data sets like the ones you are referring to this would be a valuable extension and can be easily incorporated in the workflow once the residuals are derived. However, based on our experience, we advocate that visual inspection should be included in the quality check workflow as well. If the data sets are so large that this is not feasible for all hydrographs, the ranking discussed earlier is helpful to focus the visual inspection on the most promising candidates (please see the last paragraph of the conclusion).

In general we think that it is of interest for the data quality management to have a ranking which wells exhibit relatively normal behaviour and which wells are more likely to exhibit anthropogenic effects or measurement errors. We think this holds especially for larger data sets like the ones you are referring to.

To clarify the raised issue we changed the sentence in L 157 to:

"All 141 groundwater head series were quality checked before by the LUNG. The analysed data set comprised only those time series which were considered to be without any direct anthropogenic effects or measurement errors."

Furthermore, we rephrased lines 233-236, moved them to the first paragraph of section 3.3 and rewrote lines 236-239 as second paragraph:

"In contrast to many other approaches, it is not required that the residuals are white noise or alike. Systematic structures in the residuals like drifts, trends, cyclic patterns, sudden shifts or distinct periods of deviations indicate that the respective pattern is not representative for the whole data set, but is a local peculiarity instead. This serves as indication for technical problems or anthropogenic influence.

The squared linear correlation coefficient $R^2$ of the observed versus the respective reference hydrographs was used for a first assessment which observation wells exhibit rather normal behaviour and which do not. This enabled to rank all wells in the network quickly according to their "normality". Those wells at which the observed series were not well represented by the respective reference hydrograph, i.e. those which exhibited low $R^2$ with the reference hydrograph and high $R^2$ with the residuals, were considered the most promising candidates to exhibit anthropogenic effects or measurement errors. The residuals were visually examined for peculiarities. Those were than discussed with the experts from the LUNG in the context of knowledge on the local conditions and other information available for the respective observation wells to derive hypotheses on the causing drivers."

Finally, we modified the second and third sentence of the last paragraph of the conclusion to:

"In monitoring practice, ranking all wells in the network according to their "normality" can be used to distribute resources, in particular to support the decision which observation wells, i.e. which series, should be investigated in more detail. Especially for larger data sets it can be helpful to apply time series tools like filters to identify for example step changes or trends in the analysis of the residuals. Those tools can be easily incorporated in the workflow once the residuals are derived. However, based on our experience, we advocate that visual inspection should be included in the quality check workflow as well. Furthermore the residuals can be used to categorize the deviations from the normal behaviour."

**REFEREE GRC #3:** A final general comment is that the English grammar could/should be improved; this is principally due to the use of sentences which are very long, and which contain many clauses. If these can be split and simplified I'm sure that the paper will become much more readable.

**AUTHORS:** We will go carefully through the whole manuscript and improve the phrasing according to your suggestions and the suggestions provided by the Copernicus copy-editing.
* * *
*Specific review comments (SRC):*

**REFEREE SRC #1:** Some corrections to the grammar/English have been made and tracked in the attached document. The PDF was opened in Word, which is not perfect, but should allow the authors to see the suggested edits easily.

**AUTHORS:** Thanks you very much. We included those in the revised version of the manuscript.

**REFEREE SRC #2 Line 11:** "a high spatial coverage of the monitoring networks"

This doesn't sound right. Perhaps you mean a high density of monitoring points within a network? It could be read to mean that one monitoring network should cover a large area.

**AUTHORS:** We rephrased the sentence to:

"For this purpose continuous and spatially comprehensive monitoring in high spatial and temporal resolution is desired."

**REFEREE SRC #3 Lines 73-81:** This brief review of "empirical" models is somewhat unbalanced and therefore weak. It provides examples three specific methods, for which the references are quite old, and provides one reference citing the whole of data analysis, information theory and machine learning. I think a better brief review of methods that would be of more use to the reader could be written within a similar amount of space.

Here are a few examples of other / recent approaches?

https://www.sciencedirect.com/science/article/pii/S002216941930006X

https://ngwa.onlinelibrary.wiley.com/doi/abs/10.1111/j.1745-6584.2007.00382.x

https://www.sciencedirect.com/science/article/pii/S0022169416303651

**AUTHORS:** Thank you very much. We included your suggestions. The addressed lines read now:

"Another option is to fit empirical models based on the relationships between easy to obtain independent variables and the observed water level. This has been performed for example by means of multiple linear regression (Hodgson, 1978), artificial neural networks (Coulibaly et al., 2001; Coppola et al., 2003), the combination of an artificial neural network and a linear autoregressive model with exogenous input (Wunsch et al., 2018) or different other approaches from the field of artificial intelligence like adaptive neuro-fuzzy inference system, genetic programming, support vector machine or hybrid models such as wavelet-artificial intelligence models (Rajaee et al., 2019). Others applied time series models like for example the predefined impulse response function in continuous time models (Von Asmuth et al., 2008), mixed models which combine deterministic fixed effects models with statistical random effects models (Marchant et al., 2016), or the combination of different methods from the fields of exploratory data analysis, information theory and machine learning (Sahoo et al., 2017). Those data-driven approaches make efficient use of the available data and are therefore recommended as a relatively cost and time efficient way to model groundwater level in areas with limited data (Hodgson, 1978; Coulibaly et al., 2001; Coppola et al., 2003). Another benefit is that the respective models can easily be updated once new measurements or additional variables become available (Coppola et al., 2003)."

Newly added references:

Marchant, B., Mackay, J. and Bloomfield, J.: Quantifying uncertainty in predictions of groundwater levels using formal likelihood methods, Journal of Hydrology, 540, 699–711, doi: 10.1016/j.jhydrol.2016.06.014, 2016.

Rajaee, T., Ebrahimi, H. and Nourani, V.: A review of the artificial intelligence methods in groundwater level modeling, Journal of Hydrology, 572, 336–351, doi: 10.1016/j.jhydrol.2018.12.037, 2019.

Von Asmuth, J. R., Maas, K., Bakker, M. and Petersen, J.: Modeling Time Series of Ground Water Head Fluctuations Subjected to Multiple Stresses, Groundwater, 46, 30–40, doi: 10.1111/j.1745-6584.2007.00382.x, 2008.

**REFEREE SRC #4 Lines 125-132:** The hydrogeological structure is not clearly described. You say "several regional aquifer systems". Does this mean horizontally separated aquifers, or do you mean a

series of aquifers within a vertical stratigraphical sequence that extends across the study area. This is just not clear and should be explained more systematically, preferably with a hydrogeological map (unless it is a horizontally homogeneous, but vertically stratified system).

**AUTHORS:** The term "regional aquifer system" was an attempt to translate the german "Grundwasserleiter-Komplex".

The term describes a system of aquifers of different stratigraphy which are hydraulically connected (Manhenke et al., 2001). Those were formed during different periods of glaciation. Thus, the stratigraphy, or the sequence of stratigraphies, within one aquifer system is not everywhere the same (Manhenke et al., 2001). The water management in the region is based on those aquifer systems and not on single aquifers.

Single aquifers are horizontally separated plus the vertical stratigraphical sequences of aquifers and aquitards do not extend across the study area. For an example for one typical geological section please see Figure 6 in Manhenke et al. (2001).

We adapted the second sentence in the first paragraph of  section 2.1 , added the above explanation after that and included a new paragraph break:

"The hydrogeological structure in the area consists of several "regional aquifer systems" of Pleistocene origin that are considered to be generally hydraulically separated (Manhenke et al., 2001). The term "regional aquifer system" describes a system of aquifers of different stratigraphy which are hydraulically connected (Manhenke et al., 2001). Those were formed during different periods of glaciation. Thus, the stratigraphy, or the sequence of stratigraphies, within one aquifer system is not everywhere the same (Manhenke et al., 2001). The water management in the region is based on th aquifer systems and not on single aquifers.

Single aquifers can be horizontally as well as vertically separated. The aquifers consist mainly of sandy and gravelly sediments, ..."

**REFEREE SRC #5 Line 134:** Again, here you say "uppermost three regional aquifer systems". But the use of the word "system" could be taken to mean that you are combining a series of discrete aquifer (vertical units?). It's just difficult to understand what the system is and what the aquifers are.

**AUTHORS:**  We clarified the used terminology in our reply to your comment SRC #4 and added an explanation in the manuscript, please see our reply to your comment SRC #4.

**REFEREE SRC #6 Line 140**: Can you summarise what the recharge distribution is like, otherwise it is of not much interest to the reader. Is there a recharge gradient. Are there any spatial discontinuities in recharge that could influence the PCAs?

**AUTHORS:** There is a tendency for higher mean annual groundwater recharge in the southwestern part of the federal state. However, the PCA is performed with the z-scaled groundwater head series. Thus, differences in the spatial distribution of absolute values of groundwater recharge do not effect the results. There is no evidence for spatial discontinuities in recharge.

We rephrased lines 344-346 to clarify this issue:

"Another restriction is that PCA based on z-scaled series considers only temporal patterns of groundwater head, but ignores the absolute values. Thus, it does not allow any inferences whether

the observed groundwater head on average is higher or lower than it would be under natural conditions. Then again, the z-scaling of the series makes the analysis robust against systematic differences or discontinuities in the spatial distribution of mean groundwater recharge."

**REFEREE SRC #6 Line 141:** The sentence beginning "For the same period" has no direct relevance to the analysis, is of not much interest on its own, and could be deleted.

**AUTHORS:** We deleted the sentence as you suggested.

**REFEREE SRC #7 Line 152:** "time span from the first measurement till the last measurement of all observation wells"

This is verbose. What about "the mean length of measurement record was 19.9 years"

**AUTHORS:** Thanks, we rephrased the sentence according to your suggestion.

**REFEREE SRC #8 Line 153:** "In the last decades" is vague. What about "Since approximately 20##.." You have the data so can easily estimate this.

**AUTHORS:** We rephrased the last two sentences of the first paragraph in section 2.2., moved it in the first part of the second paragraph and continued with the exceptions from the rule:

"During the monitoring period the state office aimed to take measurements consistently on the 1$^{st}$, 8$^{th}$, 15$^{th}$ and 22$^{nd}$ day of the month. Exceptions from this rule appeared for eight series (Figure S1). Bi-weekly readings were conducted from the beginning of the monitoring period at two wells till November 2005 and at one well till December 2006. Monthly readings were conducted from the beginning of the monitoring period at two wells till November 2007, at one well till September 2001 and at another well till December 2003. At one well the reading interval was changed to monthly readings in May 2002 and continued so till the end of the monitoring period."

In addition, we changed Figure S1 such that it shows only the reading dates of the eight exceptional series. See also our reply to your comment SRC #16.

**REFEREE SRC #9 Lines 176-180:** It is not really that sensible or informative to calculate autocorrelation at lag 1, as of course, one would expect a number like 0.97. What about looking at, for each timeseries, the duration over which (de-seasonalised) levels show autocorrelation i.e. the correlogram values are greater than the error bounds. Does the degree of autocorrelation across the region show any spatial pattern that relates to the PCAs?

Given that the distance varies between the boreholes, stating the mean correlation in space is also not very informative. Why not show a variogram.

**AUTHORS:** We agree that this is not a suprising result. However, we included this  information  to provide an explanation why the "PCA results were more sensitive to changes in the selection of considered observation wells (Figure 3b) than to changes in the selection of measurement dates (Figure 3a)" (lines 313-317).

We think that for this purpose it is sufficient to document with those very simple measures the basic difference between the strength of mean correlation in the temporal domain (i.e. the

autocorrelation of the groundwater head series) versus the mean correlation in the spatial domain (i.e. between the different groundwater head series).

We emphasize that the presented study does  neither include nor require an interpretation of single PCs (lines 118, 360-363, 433-435).

Thus, the question whether the spatial patterns of autocorrelation of the groundwater head series are related to the spatial patterns of single PCs is beyond the scope of the study.

**REFEREE SRC #10 Line 210:** Regarding the sentence beginning "Thus, for the assessment". There are a number of sentences that are long and that contain the word "respectively". This is generally used poorly, making the sentences difficult to read. I would split all of these sentences into two, making each simple and clear, and stop using "respectively".

**AUTHORS:** We rephrased the sentence to:
"Thus, for the assessment of the typical regional behaviour it is crucial to use only those PCs which are rather insensitive to the specific selection of analysed observation wells, i.e. complete series, and to the specific selection of measurement dates. Those PCs were considered to be representative for the monitored region in the studied period."

We replaced "respectively" with "i.e." in Lines 22, 112, 250, 262, 364 and 442.

We deleted "respectively" in Line 257 .

We rephrased the sentence in Lines 330-333 to:
"Observation well NB-Hotel Vier Tore was excluded from the PCA because its hydrograph was known to be influenced by the lowering of groundwater level due to construction works of an underground car park in approximately 100 m distance in 1997-1998 and those of another underground car park in approximately 200 m distance in 2007-2008."

We rephrased the sentence in Lines 351-354 to:
"This should be considered for any interpretation of the reference hydrographs as well as of their local deviations. However, in this study, the stable PCs used for the calculation of the reference hydrographs turned out to be rather robust with respect to the selection of observation wells (section 4.1), suggesting that the results were not primarily determined by the local clustering of the observation wells."

We rephrased the sentence in Lines 368-369 to:
"In this case the reference hydrographs and their residuals are the same whether they are calculated from the rotated or the un-rotated PCs."
Please see also our reply to your comment SRC #14.

**REFEREE SRC #11 Line 229:** "was estimated as well specific reference hydrograph" is not grammatically correct and makes the sentence difficult to understand/read.

**AUTHORS:** We rephrased the sentence to:

"The "normal" behaviour at each observation well was estimated by a well-specific reference hydrograph. The well-specific reference hydrograph was calculated by multiple linear regression of the observed series with the time series of the scores of the stable PCs."

**REFEREE SRC #12 Line 281:** "a series of minor deviations before 1997 and another relatively strong deviation in 2011 (Figure 7b and d)". It is difficult to make this out. Can you add minor tick marks to the x axis on both Figure 6 and 7? The "relatively strong deviation" looks like it is later than 2011.

**AUTHORS:** We gave a stronger grey for the grid in the background and added minor tick marks to the x axis of both figures. We furthermore changed the colours and placement of the legend according to your suggestions in SRC #13.

[Figure]

**Figure 6: (a) Time series of hydraulic head and the reference hydrograph of well Deven. (b) Time series of residuals. Location of the well is marked as black dot in the inset map. Height of the well head at well Deven is 61.48 m a.s.l.. Correlation of the observed series with the reference hydrograph and with the residuals is given as R$^2$.**

[Figure]

**Figure 7 Time series of hydraulic head and the reference hydrograph of (a) well Neubrandenburg UP and (b) well NB-Hotel Vier Tore. (c) Time series of residuals. Location of the wells is marked as black dot in the inset map. Height of the well head at well Neubrandenburg UP is 18.3 m a.s.l., at well NB-Hotel Vier Tore 17.86 m a.s.l.. Correlation of the observed series with the reference hydrograph and with the residuals is given as $R^2$.**

**REFEREE SRC #13 Figure 6&7:** The use of colour is not great, and will be undiscernible for those with (common) redgreen colour blindness. Why not use dotted and dashed lines, and please add a legend to each plot, which will make it quicker to differentiate between the reference and observed hydrographs. I think the eight lines in the very top right of the figure should be moved to their relevant position in each plot (box)

**AUTHORS:** We changed the figures according to your suggestions. Please see our reply to SRC #12.

**REFEREE SRC #14 Line 368:** "Than the reference hydrographs, respectively the residuals, are the same whether they are calculated from the rotated or the un-rotated PCs." This a poorly constructed sentence and therefore difficult to understand.

**AUTHORS:** We rephrased the sentence to:
"For the suggested screening application rotation of the PCs does not change the results as long as the rotation is performed with all stable PCs or only with a subset of the stable PCs. In this case the

reference hydrographs and their residuals are the same whether they are calculated from the rotated or the un-rotated PCs."

**REFEREE SRC #15 Lines 419-428:** The first part of the conclusion is just repetition of the methodology, and should be deleted, as it is not of mush interest here. There is generally quite a lot of repetition throughout the manuscript relating to the methodology, for example the first sentence of section 5.2.

**AUTHORS:** We deleted the first paragraph of the conclusion except the last sentence. This sentence we moved to the end of the former second paragraph, which reads now:
"Other suggested applications for the stable PCs are for example data-driven gap-filling in the observed series, spatial and temporal extrapolation of the reference hydrographs or the identification of distinct reference observation wells. The approach to determine the stable PCs is transparent and reproducible."

We rephrased the first two sentences of section 5.2 to:
"The well-specific peculiarities in the time series of the residuals of the reference hydrographs might be of different origin."

**REFEREE SRC #16 Figure S1:** This figure doesn't provide the reader with much (if any) useful information/knowledge about the dataset. Much better would be one/some plots summarising some of the statistical properties of the observations e.g. density plots summarising changes in the frequency of measurement over time would provide more useful information.

**AUTHORS:** We changed the figure such that we show now only the eight series which exhibit marked deviations from the quasi-weekly sampling interval, see also our reply to SRC #8.

[Figure]

**Figure S1 Dates of reading of groundwater head of the eight series which exhibited marked deviations from the quasi-weekly reading interval. Daily reading intervals appear as black solid line.**

**Anonymous Referee 2**

**AUTHORS:** We thank Referee 2 for her/his time, work and the constructive comments. This helped to improve and clarify our manuscript. For the review process we divided the comments in subitems.

Unaltered responses from the 3$^{rd}$ of October 2019 are marked with blue font color. New or modified responses are marked with red font color.
* * *
**REFEREE #1:** The paper addresses time series analyses for piezometric heads measured on different wells belonging to a dense network. The topic of the paper is suitable for HESS. The analyses are performed with a known statistical method (Principal Component). The novelty of the paper is the application of this method to piezometric heads chronicle to detect peculiarities in hydrographs of groundwater head.

**AUTHORS:** We thank the Referee for the generally positive valuation of our study.

**REFEREE #2:** 141 groundwater head timeseries were selected from 583 wells. The selection criteria are for me unclear. Were they selected because they reach different aquifers? Using a first screening using statistical criteria?

**AUTHORS:** The suggested approach requires that the analysed series exhibit, at least after preprocessing, the same dates (lines 83+84, 434-436). The 583 wells were not monitored all at the same time (line 151). Therefore, we chose a period during which the number of continuously monitored wells was preferably large, and during which the gaps in the series were preferably small.

Based on expert knowledge the state agency (LUNG) had performed a pre-selection to exclude wells affected by direct anthropogenic impacts (please see also our reply to comment GRC #2 of Referee 1).

To clarify this, we rephrased the sentence in line 156 to:

"For this study, we selected those 141 wells which were continuously monitored during the 20 years period from 1993-11-01 to 2013-10-22 (Figure 1)."

and the sentence in L 157 to:

"All 141 groundwater head series were quality checked before by the LUNG. The analysed data set comprised only those time series which were considered to be without any direct anthropogenic effects or measurement errors."

In the following lines, we provide now also a more detailed description on the consistency of the sampling interval during the analysed period (please see our reply to comment SRC #8 of Referee 1).

**REFEREE #3:** Furthermore, piezometric fluctuations with known anthropogenic influences are excluded from the PCA. Does it mean that the method detects only "minor" (not obvious from the visual inspection) peculiarities?

**AUTHORS:** No, the method is not restricted to "minor" peculiarities which are not obvious from the visual inspection. Please see also our reply to your comment #6.

**REFEREE #4:** Groundwater heads are usually depending on groundwater recharge, the thickness of the unsaturated zone, exchange with rivers that can have different time characteristics. Under such very different conditions, the computation of the reference hydrograph is not obvious and need some more details (see §3.3). Could you provide some more details for two very different time series and how the PCs included in the calculation are chosen?

**AUTHORS:** For each single observed hydrograph the respective reference hydrograph is calculated individually by multiple linear regression with the same stable PCs (in this study the first four PCs) (section 3.3). Thus, the regression coefficients are site specific, but not the selection of the PCs used in the regression. The stable PCs were shown to be broadly insensitive to the selection of single wells or of single sampling dates. Thus, it can be concluded that the stable PCs depict general features of groundwater head dynamics in the region. Local features which appear only at a small subset of the wells will be not assigned to the stable PCs. Consequently, local effects will be assigned to the residuals of the affected wells rather than to their reference hydrographs. For example, if there would be an influence of a river on the observed groundwater head dynamics only at a small subset of the wells than this effect would be assigned to the residuals of the affected wells.

**REFEREE #5:** Mean depth to the ground surface are analyzed. Therefore, systematic errors due to the vertical coordinate of the well reference cannot be detected.

**AUTHORS:** ~~We are not quite sure what you mean. Data about mean depth to groundwater is provided as background information for the data set. But these data were neither required nor used for subsequent analysis. However, the skewed distribution of mean depth to groundwater affects the estimation of the "normal behaviour" with the reference hydrograph. In this study it is distorted towards lower depths, as well as it is distorted towards areas with higher density of observation wells (line 350).~~
Data about mean depth to groundwater is provided as background information for the data set. But this information was neither required nor used for the PCA itself or subsequent analysis. However, the PCA results naturally refer to the analysed data. In this study, the mean depth to groundwater exhibited a skewed distribution in the monitoring network. Consequently, what is considered to be "normal behaviour" by means of the reference hydrograph refers to this skewed distribution.

We changed the wording from "biased" to "distorted" and extended the statement in line 350 to clarify this. It reads now:

"Because all the series were equally weighted by z-scaling (section 3.1), the derived PCs and consequently the determined normal behaviour was  distorted towards areas with higher density of observation wells (Karl et al., 1982) as well as towards  smaller mean depths to groundwater (Figure 2)."

**REFEREE #6:** Moreover, the provided examples show time limited peculiarities. Is the method suitable to detect long term peculiarities like drifts?

**AUTHORS:** Yes it is, as long as the drift is limited to no more than a small subset of the monitoring network. In that case the drift will  not be included in the stable PCs which are used to calculate the reference hydrographs. Instead the drift will be assigned to the residuals of the respective well(s).

This is also why for this application  the residuals do not need to fulfil specific conditions like  white noise distribution, etc. Please see lines 233-236 in the manuscript. We rephrased it according to your comment and the comment GRC #2 of Referee 1. It reads now:

"Systematic structures in the residuals like drifts, trends, cyclic patterns, sudden shifts or distinct periods of deviations indicate that the respective pattern is not representative for the whole data set, but is a local peculiarity instead."

To clarify this we extended the sentence in lines 433 + 434 of the conclusion to:

"The assignment of local anomalies to the residuals is not restricted to specific types of temporal patterns. The residuals merely comprise what cannot be ascribed to the reference hydrographs by means of the stable PCs. This can be short term structures like sudden shifts or distinct periods of deviation as well as long term structures like drifts, trends or cyclic patterns. The presented approach also does not require an interpretation of single PCs as distinct physical processes or functional relationships."

**REFEREE #7:** L319 - Reference to Kaiser criterion is 3.2 and not 3.1.

**AUTHORS:** Thank you. We corrected this.

**Author comment 1**

Published as AC2, 'Two minor changes', 9[th] September 2019:
* * *
**AUTHORS:** During the work on the comments of Referee 1 we discovered two minor issues we would like to change.

AC #1 We changed the wording in Line 173 from "biased" to "skewed":

"The distribution of monitored mean depths to groundwater was heavily skewed towards smaller depths (Figure 2)."

AC #2 We rephrased the first two paragraphs of section 5.1:

"To select only those PCs which are representative for the monitored region in the analysed period a series of PCAs were performed based on randomly selected subsets of the complete data set to identify the stable PCs. The stable PCs are those  of which the assigned spatial patterns (loadings) and temporal patterns (scores) were rather insensitive to the selection of analysed observation wells and measurement dates. Only the stable PCs were considered for the further analysis.

Earlier studies which used PCA to summarize hydrological variability in a region analysed the stability of their results in a similar way (Smirnov, 1973; Lins 1985a). Those attempts were limited to the comparison of a few PCAs, respectively a few different configurations of the data set. The correlation analysis in this study extended the assessment of stability of PCs towards random selections of the analysed data in both the spatial and the temporal domain, and a substantially larger number of different configurations of the analysed data set. "

**Author comment 2**

Published as AC5, 'One minor change', 3[rd] October 2019:
* * *
**AUTHORS:** During the work on the comments of Referee 2 we discovered one detail we would like to change to improve the clarity of the used terms at the beginning of the manuscript. In the introduction in lines 108+109 we extended the wording similar to lines 18+19 in the abstract:

[revised manuscript text omitted]

5 eight series which exhibited marked deviations from the quasi-weekly reading interval. Daily reading intervals appear as black solid line.

[Figure]

**Figure S2 Red: Boxplots of the number of days between subsequent readings in the 141 observed groundwater head series before the interpolation. Black circles: Outliers. Blue solid line: mean of the maxima of measurement intervals of all series. Magenta solid line: mean of the mean of measurement intervals of all series.**